# Improvement of Impaired Motor Functions by Human Dental Exfoliated Deciduous Teeth Stem Cell-Derived Factors in a Rat Model of Parkinson’s Disease

**DOI:** 10.3390/ijms21113807

**Published:** 2020-05-27

**Authors:** Yong-Ren Chen, Pei-Lun Lai, Yueh Chien, Po-Hui Lee, Ying-Hsiu Lai, Hsin-I Ma, Chia-Yang Shiau, Kuo-Chuan Wang

**Affiliations:** 1Division of Neurosurgery, Department of Surgery, National Taiwan University Hospital, Taipei 100, Taiwan; tefu.chen@caire.com.tw; 2Graduate Institute of Medical Sciences, National Defense Medical Center, Taipei 114, Taiwan; 3Non-Invasive Cancer Therapy Research Institute, Taipei 104, Taiwan; pohuilee@outlook.com; 4Genomics Research Center, Academia Sinica, Taipei 11529, Taiwan; d01b48001@ntu.edu.tw; 5Cancer Progression Research Center, National Yang-Ming University, Taipei 11221, Taiwan; g39005005@gmail.com; 6Department of Medical Research, Taipei Veterans General Hospital, Taipei 11217, Taiwan; d49405004@gmail.com; 7Department of Neurosurgery, Tri-Service General Hospital, Taipei 115, Taiwan; uf004693@mail2000.com.tw; 8Department of Surgery, National Defense Medical Center, Taipei 115, Taiwan; 9Graduate Institute of Life Sciences, National Defense Medical Center, Taipei 114, Taiwan

**Keywords:** human exfoliated deciduous teeth, stem cell, Parkinson’s disease (PD), rotenone

## Abstract

Parkinson’s disease (PD) is a long-term degenerative disease of the central nervous system (CNS) that primarily affects the motor system. So far there is no effective treatment for PD, only some drugs, surgery, and comprehensive treatment can alleviate the symptoms of PD. Stem cells derived from human exfoliated deciduous teeth (SHED), mesenchymal stem cells derived from dental pulp, may have promising potential in regenerative medicine. In this study, we examine the therapeutic effect of SHED-derived conditioned medium (SHED-CM) in a rotenone-induced PD rat model. Intravenous administration of SHED-CM generated by standardized procedures significantly improved the PD symptoms accompanied with increased tyrosine hydroxylase amounts in the striatum, and decreased α-synuclein levels in both the nigra and striatum, from rotenone-treated rats. In addition, this SHED-CM treatment decreased both Iba-1 and CD4 levels in these brain areas. Gene ontology analysis indicated that the biological process of genes affected by SHED-CM was primarily implicated in neurodevelopment and nerve regeneration. The major constituents of SHED-CM included insulin-like growth factor binding protein-6 (IGFBP-6), tissue inhibitor of metalloproteinase (TIMP)-2, TIMP-1, and transforming growth factor β1 (TGF-β1). RNA-sequencing (RNA-seq) and Ingenuity Pathway Analysis (IPA) revealed that these factors may ameliorate PD symptoms through modulating the cholinergic synapses, calcium signaling pathways, serotoninergic synapses, and axon guidance. In conclusion, our data indicate that SHED-CM contains active constituents that may have promising efficacy to alleviate PD.

## 1. Introduction

Parkinson’s disease (PD), the second most common neurodegenerative disease after Alzheimer’s disease, is characterized by the gradual loss of dopaminergic (DA) neurons in the substantia nigra [1,2]. The pathogenesis of PD remains poorly understood, but the causes that lead to PD have been linked to environmental and genetic factors, oxidative stresses, mitochondrial dysfunction, and neuroinflammation [3]. Significant PD symptoms, including tremors, bradykinesia, and stiffness, can be initially detected in patients with approximately 60% of nigral dopaminergic neuron degradation and 80% of striatum dopamine loss, therefore making the early diagnosis of PD extremely difficult [4]. PD patients may suffer from cognitive retardation and depression as the disease progresses, and the cognitive functions of many patients can be reduced by about 80% [3,5]. Cell replacement therapy using embryonic stem cells (ESCs) [6], mesenchymal stem cells (MSCs) [7,8], or neural stem cells [9] may provide therapeutic opportunity for neurodegenerative diseases. Although accumulated data revealed the therapeutic potential of cell therapy using these cell types in experimental models, it is still argued that cell therapy using ESCs usually carries ethical issues and tumorigenic risks. In addition, cell therapy using ESCs does not always exhibit long-term efficacy, and severe side effects such as graft-induced dyskinesia have been reported in PD patients after ESC transplantation [10,11]. MSCs can be found within many tissues, including the bone marrow, adipose tissue, and the umbilical cord. Stem cells from human exfoliated deciduous teeth (SHED), a type of well-defined MSCs, are thought to originate from the cranial neural crest and express early MSCs and neuroectodermal stem cell markers [12]. Previous studies have shown that SHED can differentiate into oligodendrocytes and neurons under defined conditions [13,14,15] and have immunomodulatory and regenerative activities. Transplantation of SHED into rats with spinal cord injury may promote the recovery of motor function through a paracrine mechanism [15,16,17]. Recent evidence showed that administration of the conditioned medium of SHED (SHED-CM) could alleviate the severity of experimental autoimmune encephalomyelitis [18] and cognitive function in a model of Alzheimer’s disease [19]. It is plausible that the efficacy can be attributed to the bioactive constituents secreted from SHED [20] and systemic administration SHED-CM offers those anti-inflammatory effects could result from converting the microglia into M2 phenotype [21]. Interestingly, whether the bioactive constituents in SHED-CM also exhibit therapeutic effects that can ameliorate PD symptoms remains uncertain.

Rotenone is a naturally occurring isoflavone compound used to induce a highly reproducible PD model. Such a rotenone-induced PD model offers many advantages over other experimental PD models and highly replicates the phenotypes and the neuropathological features of human PD through selective degeneration of dopaminergic neurons [22]. In the present study, we investigate the therapeutic potential of SHED-CM in a rotenone-induced PD rat model.

## 2. Results

### 2.1. SHED-CM Rescued the Motor Deficits in Rotenone-Induced PD Rat

Rotenone is an extensively used chemical for the induction of PD in experimental models [22]. Here, we induced PD via the intraperitoneal injection of rotenone at the dose of 2.5 mg/kg/day for 10 consecutive days. After induction, these rats showed a mild decrease in body weight (Figure 1A). As examined by the Rotarod test, a well-known system to test brain function, these rotenone-treated rats also exhibited significant posture instability and behavioral defects and simultaneously lost the ability to travel for a longer period on the Rotarod device (Figure 1B). The brain from rotenone-treated rats was then subjected to immunohistochemistry analysis for various PD-specific markers. Importantly, ventriculomegaly with remarkable shrinkage of the whole striatum and severe hydrocephalus was noted in rotenone-treated rats (Figure 1C). These phenotypes and pathological changes validated the induction of PD. We next collected SHED-CM with standardized protocols and examined the treatment efficacy of SHED-CM in rotenone-treated PD rats. SHED-CM was injected via an intravenous route at either 10 μg/mL, 30 μg/mL, 100 μg/mL or 400 μg/mL. We evaluated and compared the behavioral performance among normal control rats, rats with SHED-CM injection, rotenone-treated PD rats, and rotenone-treated PD rats with SHED-CM injection. SHED-CM did not show any observable effect on behavioral performance in control rats (Figure 1D). It was determined that 10 μg/mL of SHED-CM failed to restore the motor ability of PD rats, 30 μg/mL of SHED-CM mildly restored the ability of PD rats to move on the Rotarod roller, 100 μg/mL of SHED-CM led to the maximal improvement of motor deficits of PD rats, while 400 μg/mL of SHED-CM treatment did not reveal further improvement than the 100 μg/mL treatment group (data not shown). The dose of 100 μg/mL was therefore used as the treatment dose for subsequent experiments.

### 2.2. SHED-CM Ameliorated the Pathological Features in the Brain from Rotenone-Induced PD Model

We next evaluated the pathological features in the rotenone-treated PD model and examined the treatment effect of SHED-CM in such a model. After the induction of PD and indicated treatment, striatum and substantia nigra were isolated and collected for analyses. In the immunohistochemistry analysis of striatum, rotenone treatment led to severe shrinkage of striatum accompanied with decreased expression of tyrosine hydroxylase (TH; Figure 2A,B) and simultaneously increased the accumulation of α-synuclein (a-syn; Figure 2E,F). Remarkably, the administration of SHED-CM significantly restored TH expression (Figure 2A,B) and suppressed the accumulation of a-syn in the striatum (Figure 2E,F). In the isolated substantia nigra, rotenone significantly decreased expression of TH (Figure 2C,D) and increased the accumulation of a-syn, and SHED-CM restored TH expression (Figure 2C,D) and led to a suppressive effect on this rotenone-induced a-syn accumulation (Figure 2G,H). Along with the observations of behavioral deficits and pathological features, SHED-CM exhibited a remarkable efficacy that could ameliorate the severity of PD in the rotenone-induced experimental model.

### 2.3. SHED-CM Significantly Ameliorated Neuroinflammation in Different Brain Areas from Rotenone Induced PD Rats

Diffuse neuroinflammation is another characteristic of the brain in the PD experimental model. We next examined the treatment effect of SHED-CM on inflammation in the rotenone-induced PD model. Iba-1 is a marker of activated microglia and can be used for examining the involvement of neuroinflammation. Immunohistochemistry revealed that rotenone led to a substantial increase of Iba-1-positive cells in the striatum, substantia nigra, and the cortex, indicating diffuse microglial activation and neuroinflammation (Figure 3A). Remarkably, the intravenous administration of SHED-CM significantly reduced the amounts of Iba-1-positive cells in these brain areas in rotenone-induced PD rats (Figure 3A–D). CD4-positive T cell accumulation has been shown as another feature of neuroinflammation and subsequent neurodegeneration [23]. Immunohistochemistry staining also revealed increased amounts of CD4-positive cells in both nigra and striatum in rotenone-induced PD rats (Figure 4A–C). Administration of SHED-CM similarly reduced the amounts of CD4-positive cells in these brain areas (Figure 4A–C). Taken together, our data showed that SHED-CM treatment exhibited therapeutic potential that could effectively ameliorate neuroinflammation in the rotenone-induced experimental model of PD.

### 2.4. SHED-CM Treatment Normalized the Global Gene Profiles in Rotenone-Induced PD Rats 

To investigate whether SHED-CM could rescue neuronal degeneration in the brain of PD rats, RNA sequencing was performed to identify genes differentially expressed in rotenone-induced PD rats treated with or without SHED-CM, compared to control untreated rats (Figure 5A). The global gene expression profiles revealed that the SHED-CM administration shifted the gene expression profile to a pattern similar to that of control untreated rats, compared to that of the rotenone-treated PD group. Gene Ontology analysis revealed that SHED-CM significantly upregulated the genes that are involved in neurodevelopment and nerve regeneration, including those genes responsible for the regulation of axon extension and axon guidance, potassium ion transmembrane transports, etc. (Figure 5B). These bioinformatics results indicated that SHED-CM is capable of promoting the neural regeneration and shifting the PD-specific gene expression profiles to a pattern comparable to that in control untreated rats at the molecular levels.

### 2.5. The Major Constituents in the SHED-CM

To further determine the major constituents in the SHED-CM that contribute to the improvement of PD, we used ELISA array to measure the levels of cytokines/growth factors contained in the SHED-CM. Notably, the constituents in the SHED-CM with high levels included insulin-like growth factor binding protein 6 (IGFBP-6) (29.46%), tissue inhibitor of metalloproteinase 2 (TIMP-2) (26.42%), tissue inhibitor of metalloproteinase 1 (TIMP-1) (17.97%), transforming growth factor-beta 1 (TGF-β1) (8.06%), insulin-like growth factor binding protein 2 (IGFBP-2) (6.02%), insulin-like growth factor binding protein 4 (IGFBP-4) (5.48%), bone morphogenetic protein 5 (BMP-5) (4.48%), and others (2%) (Figure 6A). Furthermore, we performed RNA-sequencing (RNA-seq) and Ingenuity Pathway Analysis (IPA) to analyze these major constituents in combination with the transcriptomic data and elucidated the possible mechanisms/pathways that were responsible for the SHED-CM-mediated improvement of PD symptoms (Figure 6B). The predicted mechanisms/pathways involved in the SHED-CM treatment effect are illustrated according to the results of ELISA and RNA-seq (Figure 6B). The networks were further grouped according to the candidate mediators with abundant expression, showing that the mechanisms by which SHED-CM ameliorated PD may include cholinergic synapses, calcium signaling pathways, serotoninergic synapses, and axon guidance (Figure 6C).

## 3. Discussion

In this study, we collected the conditioned medium secreted by stem cells from human exfoliated deciduous teeth (SHED-CM) and sought to examine the treatment effect of SHED-CM to rescue the symptoms and neurological deficits in the rotenone-induced PD experimental model. Our findings demonstrated that SHED-CM generated by standardized and defined procedures exhibited remarkable anti-inflammatory effects on rotenone-induced PD-like symptoms, pathology, and transcriptomic changes (Figure 7). The SHED-CM-mediated mechanisms may include the upregulation of TH expression, the reduction in α-synuclein accumulation (Figure 2), and decreased amounts of Iba-1-positive cells (Figure 3) and CD4-positive cells (Figure 4). Gene Ontology analysis indicated that SHED-CM led to the upregulation of the genes involved in neurodevelopment and nerve regeneration (Figure 5), and the major constituents of SHED-CM may participate in the molecular networks related to cholinergic synapses, calcium signaling pathways, serotoninergic synapses, and axon guidance (Figure 6). Injection of SHED-CM generated by standardized procedures significantly improved the phenotype of PD and restored the motor activity defects and dopaminergic neuron loss caused by rotenone. Collectively, the secretory factors derived from SHED may possess promising therapeutic potential in the treatment of PD.

As analyzed by Drago et al., the average molecular weight of these constituents was found to range from 7 to 85 kDa [24]. Among active constituents of CM are vascular endothelial growth factor (VEGF), hepatocyte growth factor (HGF), brain-derived neurotrophic factor (BDNF), glial-derived neurotrophic factor (GDNF), stromal-derived factor-1 (SDF-1), fibroblast growth factor (FGF), IGF-1, TIMP-2, TIMP-1, and TGF-β1[24]. Our fractionation and purification work for bioactive constituents using tangential flow filtration (TFF) in combination with centrifugation and filtered via a 0.22 mm filter is a well-accepted procedure for protein purification [25,26]. In addition, filtration with 3 or 5 kDa filter cut-off is also a routine step for removing protease, peptidase, or other small compounds. In our experimental setup, particles with molecular weights larger than 30 kDa were considered as cell debris or impurities in the conditioned medium. Therefore, we removed large particles >30 kDa to avoid non-specific effects and to allow the active constituents in conditioned medium to pass the blood–brain barrier[27].

The treatment efficacy of mesenchymal stem cell-derived conditioned medium has been examined in a variety of disease models or conditions, including hair loss [28], myocardial infarct [29], cerebral injury [16,30], ischemia [31], spinal cord injury [32], lung injury [33], bone defect [34], wound therapy [35], and liver injury [36]. In this study, we fractionated the proteins within the 5–30 kDa range from the SHED-CM following our established protocol [27], and further used ELISA array to examine the constituents within this fraction. The major constituents within the 5–30 kDa fraction were found to be IGFBP-6, TIMP-2 and -1, and TGF-β1. RNA-seq and Ingenuity Pathway Analysis were used to further predict the molecular networks mediated by these major constituents based upon the results of the ELISA array (Figure 6). SHED cells and dental pulp stem cells are an accessible source for stem cell therapy for PD [37,38]. SHED originate from the cranial neural crest and express MSC markers just like MSCs derived from the adipose tissue or Wharton’s jelly. SHED has been reported to exhibit capability of diminishing PD phenomena caused by rotenone [39] and the exosomes from SHED was shown to retard apoptosis of the neuro-progenitor cells in an in vitro PD model caused by 6-hydroxydopamine (6-OHDA) [40]. Moreover, the SHED-CM normalized the TH in a 6-OHDA induced PD rat animal model [41]. Here, we tried to prove the treatment potential in rotenone-induced PD rats with SHED-CM and disclosed the potential mechanisms with the aid of transcriptomes ontology simulation.

Our data revealed IGFBP-6 as the most abundant secretory protein among the constituents in SHED-CM. Previously, we demonstrated that the conditioned medium from the culture of dental pulp-derived stem cells could alleviate neuroinflammation via an IGF-1-related mechanism in a rat model of subarachnoid hemorrhage [27]. Interestingly, IGFBP-6 was reported to be an important neuronal survival factor that can inhibit IGF-2-mediated apoptotic effect and promote IGF-1-mediated neuroprotection [42]. IGF-1 has been reported as a key growth factor that regulates neurogenesis and synaptogenesis as well as the generation, differentiation, and maturation of neurons [43], and exerts wide-spectrum neuroprotection against neuroinflammation and oxidative stress in the brain [44]. Therefore, it is likely that IGFBP-6 provides neuroprotective effects via an IGF-1-dependent mechanism to reduce the activated microglia with Iba-1-positive staining and CD4-positive T cells in rotenone-treated rats. In addition to IGFBP-6, TIMP-2 was the second most abundant factor among those identified constituents and has been reported to rejuvenate the hippocampus, promote nerve regeneration, and improve cognitive functions in aged rats [45]. TIMP-1 has been shown to serve as a crucial modulator of neuronal outgrowth and morphology through inhibition of matrix metalloprotease-2 in a paracrine or autocrine manner [46]. TGF-β1 and IGFBP-2 accounted for the minor proportion among the constituents in the SHED-CM. TGF-β1 has multiple functions, including the inhibition of chemokines/adhesion molecules involved in the regulation of central nervous system (CNS) leukocyte trafficking, microglia function, the phagocytic ability of pericytes, and the stimulation of typical proinflammatory cytokines. Overall, TGF-β1 is believed to exhibit important anti-inflammatory and regenerative effects on brain damage [30,47,48]. For IGFBP-2, it has been reported to stimulate the proliferation and differentiation of neural stem cells [49]. Together, these factors may mediate the neuroprotective mechanisms predicted according to the results of ELISA and RNA-seq (Figure 6B,C).

α-Synuclein has been linked genetically and neuropathologically to PD. The accumulation of α-synuclein in axons [50] and the presynaptic terminal [51] was shown to contribute to neurodegeneration in PD [50] and dementia [51]. Along with the observations of previous studies [52,53], our data also support that α-synuclein accumulations serve as the triggers of onset and formation of PD-related synaptopathy. Our immunohistochemistry results showed that SHED-CM treatment significantly diminished α-synuclein accumulation in many brain areas (Figure 2). Mitochondria is not only a cellular energy factory but also an important regulator in calcium homeostasis and signal transduction [54,55]. Importantly, mitochondrial dysfunction has been reported as a major factor underlying PD pathogenesis [3], and calcium homeostasis may play an important role in endoplasmic retinaculum–mitochondrial interaction in PD [56]. In the present study, Gene Ontology results indicated that potassium ion transmembrane transport was the second most upregulated pathway involved in the SHED-CM-mediated neuroprotection (Figure 5B). The IPA also showed that the major constituents in the SHED-CM are associated with the regulation of calcium signaling pathways (Figure 6C). Interestingly, potassium channels, such as the ATP-regulated potassium channel and the calcium-activated potassium channel, are present in the inner mitochondrial membrane [57]. Collectively, our data provided the plausible neuroprotective mechanisms of SHED-CM that are associated with the recovery of mitochondrial functions and the regulation of the potassium and calcium ion transmembrane transport genes. However, all those speculated mechanisms derived from SHED-CM treatment in this PD rat animal model should undergo further studies in the future.

In the present study, our findings demonstrated that the SHED-CM generated by our established and standardized procedures showed a prominent therapeutic efficacy that alleviated intracerebral CD4-positive cells, microglia activation, and α-synuclein accumulation, in brain areas (especially the substantial nigra and striatum), and concurrently improved the neurological deficits in a rotenone-induced PD rat model. MSCs have extensively demonstrated promising potentials in central nervous system (CNS) diseases such as PD [58,59] and dementia [58,59]. Remarkably, the conditioned medium from MSCs (MSC-CM) has also been proven to exhibit effective therapeutic benefits against CNS diseases, including focal cerebral ischemia-reperfusion injury [60], neural trauma, and stroke [61]. Although the efficacy of MSC-CM reported in these studies was generally prominent, the mechanisms that MSC-CM contributed to its therapeutic effect remain unclear. Although most studies reported that MSC-CM showed promising therapeutic potential predominantly when administered via an intracerebral route [60], some studies also demonstrated that peripheral administration of MSC-CM also exhibited dramatic efficacy to ameliorate CNS damage in stroke and neural trauma [61], and ischemia-reperfusion injury in the brain [62,63,64]. Consistent with these observations of MSC-CM effect via peripheral administration [61,62,63,64], our data also revealed that intravenous injection of MSC-CM alleviated neural damage in the CNS injured by rotenone. 

In the current study, we speculated that there are three possibilities that contributed to the MSC-CM-mediated therapeutic effects. First, some of small-sized proteins might be able to penetrate the blood–brain barrier (BBB) in a normal status [65]. Therefore, it is plausible that certain or some active constituents in the MSC-CM could penetrate the BBB to rescue the brain damage induced by rotenone. Hyperpermeability and breakdown of the BBB in PD may form the underlined mechanism which allows larger molecules or drugs to pass the BBB into the CNS [66]. Moreover, this speculation might be able to partially explain the reason why normal rats that received 30 and 100 μg/mL SHED-CM treatment did not exhibit differences between the control group in motor function tests (Figure 1D). Second, some factors or cytokines (e.g., IGF-1 or TGF-β1) may stimulate the BBB to ameliorate neuroinflammation in the cerebral parenchyma [67], leading to the possibility that SHED-CM may indirectly decrease neuroinflammation via affecting the BBB. Third, it has been reported that intravenous administration of human umbilical MSCs can attenuate amphetamine-induced intracerebral inflammation by increasing interleukin-10 and forkhead box P3 (FOXP3) expression and decreasing tumor necrosis factor-α production in the serum and lymph nodes [68]. An explanation could be that peripheral administration of SHED-CM indirectly ameliorated neuroinflammation in the CNS through its immuno-modulatory abilities in which the activation of regulatory T cells and cytokines may be involved. However, further studies will be needed to elucidate the precise mechanisms by which peripheral administration of SHED-CM rescued neural damage in PD.

## 4. Materials and Methods 

### 4.1. Animals and Supplies

Animal studies were approved by the Institutional Animal Care and Use Committee (IACUC) of the National Defense Medical Center Laboratory Animal Center (NDMCLAC)-(IACUC 17-083). Eight-week-old female Lewis rats (LEW rats) (250–300 g) were purchased from BioLASCO Taiwan Co. (Taipei, Taiwan). The animals were maintained at 24 °C under a 12-h light/dark cycle (lights on 07:00–19:00). Rats were housed in standard laboratory cages and had free access to food and water throughout the study period. All animal experimental procedures and methods were performed following the relevant guidelines and regulations.

### 4.2. Rotenone-Induced Parkinson’s Disease in Rats

Thirty-six LEW rats were used in this study (including pre-testing). The rotenone was administered via an intraperitoneal route. The rotenone solution was prepared as a 50X stock with dimethyl sulfoxide (DMSO) and diluted by triglyceride, Miglyol 812 N (Sasol North America, Inc., Houston, TX, USA) to obtain a final concentration of 2.5 mg/mL rotenone in 98% Miglyol 812 N, 2% DMSO. The solution was stored in a brown vial protected from light and inverted several times before each injection. The rotenone was administered at 2.5 mg/kg and the control injected with the vehicle for 10 days to induce a PD phenotype following a previous study [69]. The body weight of these rats was monitored and then PD symptoms were tested with Rotarod running tests. 

### 4.3. Stem Cells of Human Exfoliated Deciduous Teeth (SHED) Isolation and Culture

All experimental protocols were approved by Human Subject Research Ethics, National Defense Medical College (Taipei, Taiwan) (ID: IACUC-17-083, 21 March 2017). After the milk teeth were retrieved from the donors (two nine-year-old boys and one seven-year-old boy) without extra procedures or anesthesia, SHED cells were extracted using a syringe needle from the root of the deciduous tooth and transferred into a 25 cm^2^ flask (Corning). All tissue underwent the standard protocol. The tissue was then cultured in alpha minimal essential medium (α-MEM) with 10% fetal bovine serum (FBS) (Gibco). Isolation of SHED was not subjected to any type of depletion techniques, and when they reached confluence the SHED cells were incubated at 37 °C in an atmosphere containing 5% CO_2_ at 100% humidity. The SHED used in this study exhibited a fibroblastic morphology with bipolar spindle shape, expressed MSC markers (CD90, CD73, and CD105), but not endothelial/hematopoietic markers such as CD34 and CD45. The flow cytometry data are in Appendix A.

### 4.4. Preparation of SHED-CM

The SHED-CM was generated as follows: when passage 3 SHED cells reached 80% confluence in a 150 cm^2^ cell culture flask, the medium was changed with 10 mL Dulbecco’s modified Eagle’s medium (DMEM) and 20 mL Hanks’ balanced salt solution (HBSS) in each flask, then incubated for 72 h at 37 °C in an atmosphere containing 5% CO_2_ at 100% humidity. The medium was collected and centrifuged for 3 min at 2500 rpm. In order to collect proteins with molecular weights between 5 and 30 kDa, CM was collected in a sterilized beaker, and further concentrated by Tangential Flow Filtration (TFF) membrane filter system (Millipore) with a 5 kDa and 30 kDa cut-off unit (Millipore) alternatively for 3 h following the manufacturer’s instructions. The protein concentration in SHED-CM was measured using a Pierce™ BCA Protein Assay Kit (Thermo Fisher Scientific) and adjusted to 100 ± 2.8 µg/mL with normal saline. Quantitation of each constituent was analyzed by ELISA array, Quantibody® Human Cytokine Antibody Array 4000 (RayBiotech), following the manufacturer’s instructions. This array can detect 200 target proteins, including human inflammatory factors, growth factors, chemokines, receptors, and cytokines.

### 4.5. SHED-CM Treatment

Animals were divided into 4 groups: 1) control group, sham operation; 2) control group + SHED-CM; 3) rotenone group, rotenone-induced PD; 4) rotenone group + SHED-CM. The SHED-CM was administered via tail vein intravenous injection for one-shot with/without rotenone induction. Before the formal study, pre-testing of adequate treatment dosage of SHED-CM was divided into 10 μg (10 μg/mL, 1 mL), 30 μg (30 μg/mL, 1 mL), 100 μg (100 μg/mL, 1 mL), or 400 μg (400 μg/mL, 1 mL) and there were 3 animals in each group. According to the results of the aforementioned pre-testing, the rats were administered 100 μg of CM intravenously for further experiments. After 2 weeks of SHED-CM treatment, all rats underwent behavior evaluation, brain histology, and brain tissue RNA-sequence analysis.

### 4.6. Behavior Evaluation by Rotarod System

The Rotarod test was performed as described by Vogel et al. [70] with small modifications. The animals were trained for three consecutive days at the speed of 5 rpm, with three sessions per day for 5 min each. If a rat fell during the habituation period, it was placed back on the instrument. On the following day, the test trial was performed. After the rats were placed on the instrument (Panlab Rota Rod, Havard Apparatus) moving at the speed of 4–40 rpm, in 600 s, the accelerating mode started (maximum speed 40 rpm). The latency to fall was measured during the 5-min test session.

### 4.7. Immunohistochemistry Staining

The immunohistochemistry (IHC) was performed by the Avidin-Biotin complex (ABC) system. Rats were sacrificed after behavior evaluation by injection of pentobarbital (200 mg/Kg, i.p.) and perfused transcardially with 50 mL of saline followed by 500 mL of a fixative containing 4% paraformaldehyde in 0.1 M phosphate-buffered saline (PBS), pH 7.3 for 30 min. Then serial brain sections, 10 μm thick, were cut on a cryostat, thaw-mounted onto gelatin-coated slides, and used for immunohistochemical staining. The samples were de-paraffinized by heating at 60 °C for 30 min and xylene. They were then rehydrated by passing through a series of decreasing concentrations of ethanol (100%, 90%, 70%, and 50%) for 5 min each step, and then washed with 0.1 M PBS. The brain section was incubated in 1% H_2_O_2_ for blocking endogenous peroxidase activity, then washed by PBS and hybridized in blocking buffer (1% goat serum, 0.5% Triton-X100) with the primary antibody, anti-α-synuclein (1:80, #AB5334P; Millipore, Billerica, MA, USA), anti-tyrosine hydroxylase (1:100, #MAB318; Millipore, Billerica, MA, USA), anti-CD4 (1:80, Serotec), and anti-Iba-1 (1:80, #019–19741; Wako) at 4 °C, overnight. Then it was washed by PBS, hybridized with biotinylated donkey anti-mouse secondary antibody (1:200, Jackson Immuno Research; West Grove, PA, USA) for 1 h then transferred to ABC solution (Vectastain, Vector Labs; Burlingame, CA, USA) for 1 h. After being washed by PBS, 0.05% 3,3′-diaminobenzidine (DAB) solution was added by shaking gently and the stain was established. Once the stain performed fully, the brain section was transferred into the PBS to stop the reaction. Finally, the sections were fixed on a slide and observed with a microscope. The photos were analyzed and quantified by Image J software (version 1.52, National Institute of Health). The IHC images were switched to grayscale and defined the grey threshold, then the gray integrated density of the staining above the threshold was measured (the definition of the threshold was the gray level value when the stained cells were completely removed, leaving only the background).

### 4.8. Tissue Isolation, RNA Extraction and Sample Quality Control 

Under intraperitoneal injection of 3% chloral hydrate (10 mL/kg), the experimental rats were sacrificed, and the brains were isolated immediately. The right and left striatum were then dissociated and rinsed with PBS, frozen in –196 °C liquid nitrogen, and kept at –196 °C until further use. Total RNA from the rat brain was extracted with RNAiso Plus reagent (Takara Bio, Kusatsu) according to the manufacturer’s instructions. From the RNA samples to the final data, each step influences the quality of the data, and data quality directly impacts the analysis results. To guarantee the reliability of the data, quality control (QC) was performed at each step of the procedure. There were three main methods of QC for RNA samples: (1) nanodrop: preliminary quantitation; (2) agarose gel electrophoresis: tests RNA degradation and potential contamination; (3) Agilent 2100: checks RNA integrity and quantitation.

### 4.9. Library Construction and Sequencing

After the sample QC procedures, a total amount of 1 μg RNA per sample was used as input material for the sample preparations and the mRNA was enriched using oligo(dT) beads. The sequencing libraries were generated using NEBNext® UltraTM RNA Library Prep Kit for Illumina® (NEB, Ipswich, MA, USA) following the manufacturer’s recommendations and index codes were added to attribute sequences to each sample. First, the mRNA was fragmented randomly by adding a fragmentation buffer, then the cDNA was synthesized by using the mRNA template and random hexamers primer, after which a custom second-strand synthesis buffer (Illumina), deoxy-ribonucleoside triphosphate (dNTPs), RNase H, and DNA polymerase I were added to initiate the second-strand synthesis. Second, after a series of terminal repair, A ligation, and sequencing adaptor ligation, the double-stranded cDNA library was completed through size selection and PCR enrichment. The libraries were pooled and run on an Illumina NovaSeq using paired-end 150 bp Rapid Run format to generate 20 million total reads per sample. 

### 4.10. Data Analysis 

Raw reads of RNA-seq from the sequencing instrument were first trimmed from the low-quality tranche and checked. The Spliced Transcripts Alignment to a Reference (STAR) software was used to map spliced short-read (RNA-seq reads) to the reference genome (Ensembl Rnor_6.0). Based on spliced alignments, transcripts reconstruction and estimation of transcripts abundance were conducted by Cuffquant. Normalized gene expressions were performed by calculating the number of RNA-seq fragments per kilobase of transcript per total million fragments mapped. To identify the differentially expressed genes, the Cuffdiff was used to tests the statistical significance of observed changes and identify genes that are differentially regulated at the transcriptional or post-transcriptional level.

### 4.11. Bioinformatics Analysis-Ingenuity Pathway Analysis (IPA)

Significant pathway enrichment analysis was performed using Ingenuity Pathway Analysis (IPA; Qiagen, Redwood City, CA, USA). IPA is a commercial tool that is based on a proprietary database to facilitate the identification of biological themes in proteomics or gene expression data. Differentially-expressed genes from the RNA expression data are associated with a biological function supported by at least one publication in the Ingenuity Pathways Knowledge Base. The identified proteins/pathway, together with the top-ranked most SHED-CM-relevant proteins are obtained. Statistically significant biological pathways were then identified by selection for pathways with Benjamini–Hochberg adjusted *p*-values < 0.05.

### 4.12. Statistical Analysis

All statistical data are presented as the mean ± standard deviation (SD) of at least three biological replicates. Statistical analysis was performed by unpaired two-tailed t-tests and one-way ANOVA with GraphPad Prism 8.2.1 (GraphPad Software, San Diego, CA, USA), where a *p*-value < 0.05 was considered a significant difference. With ordinary one-way ANOVA, the post hoc multiple comparisons tests were Tukey’s multiple comparisons test.

## 5. Conclusions

In this study, our data provided evidence showing that the SHED-CM prepared by our standardized procedures exhibits promising efficacies in the amelioration of neuroinflammation, α-synuclein clearance, the recovery of mitochondrial damage, and the improvement of motor deficits in a rotenone-induced rat model of PD. Intravenous administration of an adequate and single dose of SHED-CM is able to significantly improve the neurological outcome in PD rats, suggesting that such therapy may represent a novel potential cell-free therapy for PD treatment in the future.

## Figures and Tables

**Figure 1 ijms-21-03807-f001:**
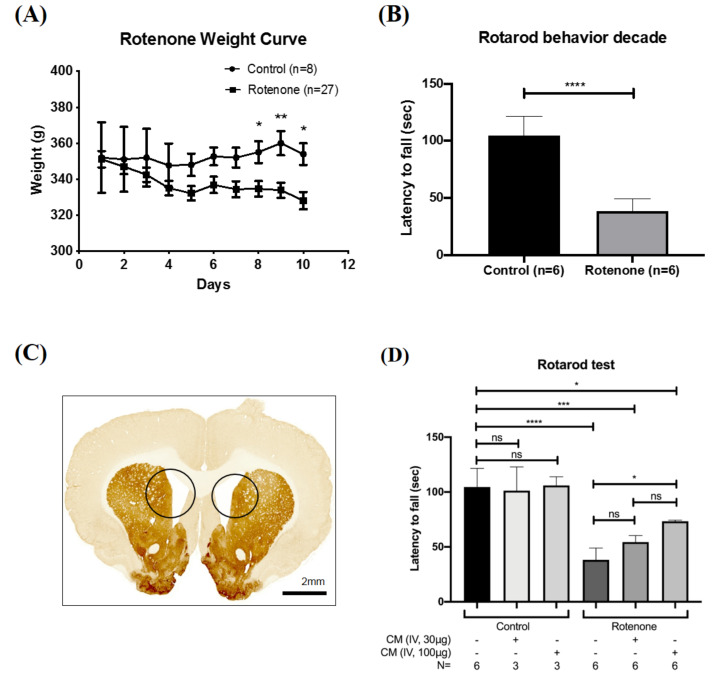
Human exfoliated deciduous teeth-derived conditioned medium (SHED-CM) ameliorates the motor deficits in the rotenone-induced Parkinson’s disease (PD) experimental model. (**A**) Rotenone administration induced a mild weight loss in rats. (**B**) Rotarod test revealed the motor deficits in rotenone-treated rats. (**C**) Remarkable shrinkage of the striatum and severe hydrocephalus were noted (black circle) in rotenone-treated rats. (**D**) Intravenous administration of SHED-CM at either 30 μg/mL or 100 μg/mL showed significant improvement of motor ability in rotenone-treated rats. * *p* < 0.05, ** *p* < 0.01, *** *p* < 0.001, **** *p* < 0.0001, ns: no significant differences.

**Figure 2 ijms-21-03807-f002:**
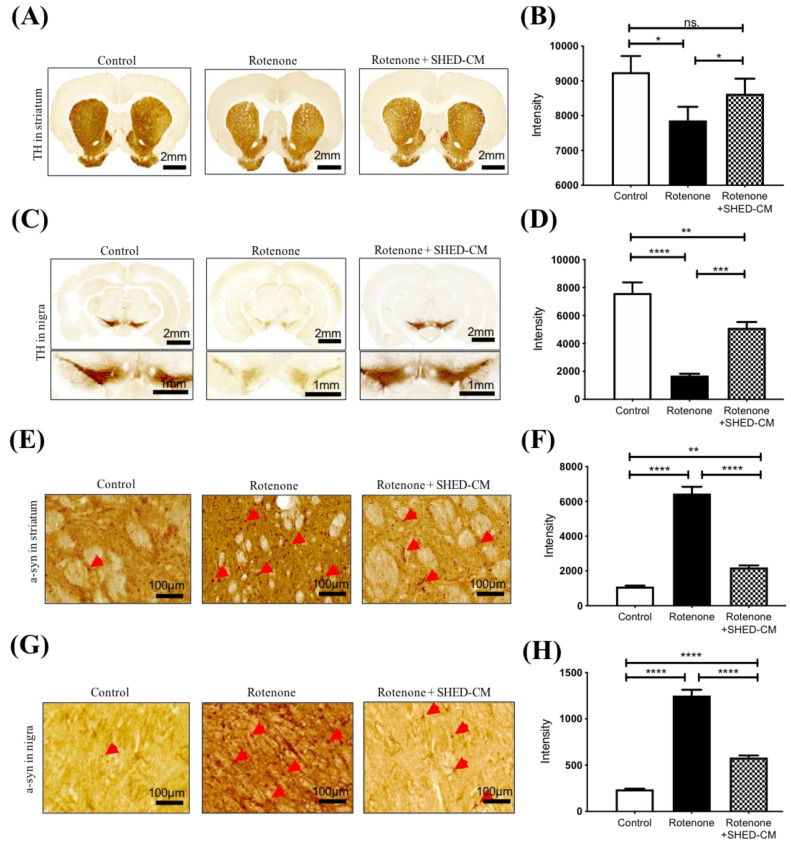
Amelioration of PD pathological features by SHED-CM in different brain areas from rotenone-induced PD rats. Immunohistochemistry showed the lateral ventricle size and tyrosine hydroxylase (TH) expression in the striatum (**A**) and substantia nigra (**C**) from rotenone-treated PD rats with indicated treatment. Quantification of TH expression in the striatum (**B**) and substantia nigra (**D**) from PD rats with indicated treatment. (**E**) Immunohistochemistry showed the α-synuclein (a-syn) accumulation in the striatum from rotenone-treated PD rats with indicated treatment. (**F**) Quantification of α-synuclein (a-syn) accumulation in the striatum from PD rats with indicated treatment. (**G**) Immunohistochemistry showed the a-syn accumulation in the substantia nigra from rotenone-treated PD rats with indicated treatment. (**H**) Quantification of a-syn accumulation in the substantia nigra from PD rats with indicated treatment. The red arrows indicate stained cells. Each group *n* = 3, * *p* < 0.05, ** *p* < 0.01, *** *p* < 0.001, **** *p* < 0.0001, ns: no significant differences.

**Figure 3 ijms-21-03807-f003:**
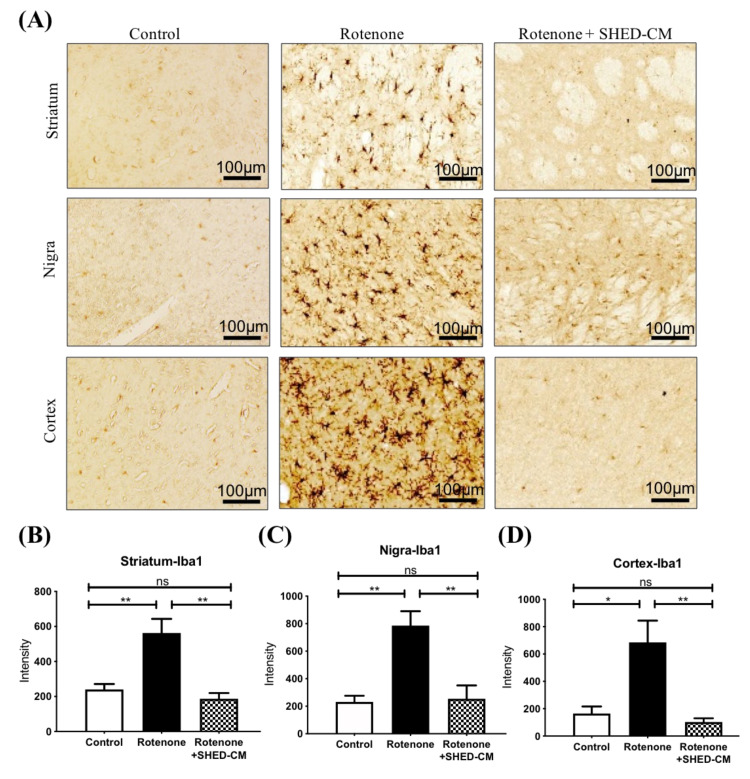
SHED-CM treatment reduced brain Iba-1 expression in rotenone-induced PD rats. (**A**) Immunohistochemistry staining of Iba-1-positive cells represented the activated microglia in the striatum, substantia nigra, and the cortex from the PD rats with indicated treatment. Quantification of the relative amount of Iba-1-positive cells in (**B**) the striatum, (**C**) substantia nigra, and (**D**) cortex. Each group *n* = 3, * *p* < 0.05, ** *p* < 0.01, ns: no significant differences.

**Figure 4 ijms-21-03807-f004:**
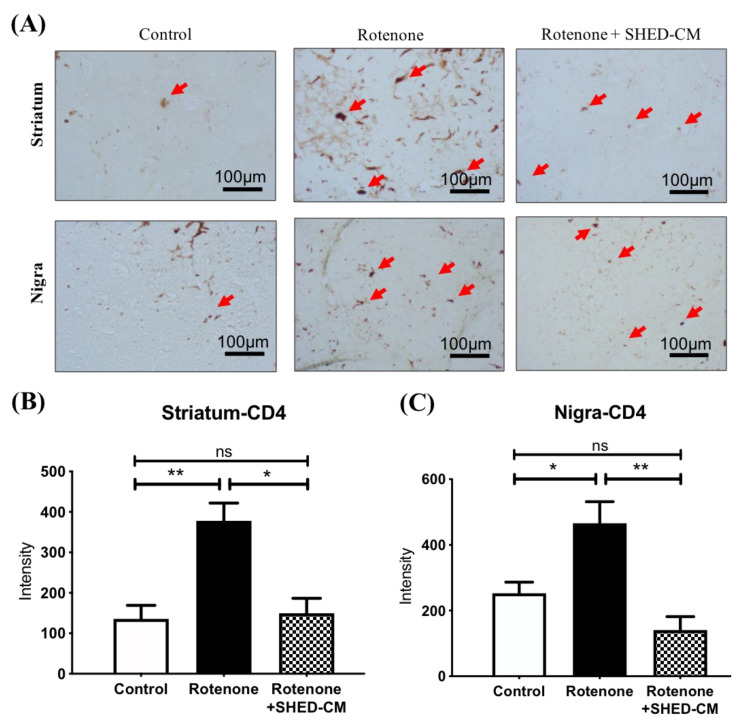
SHED-CM reduced CD4-positive T-cell migration in rotenone-induced PD rats. (**A**) Immunochemistry staining of CD4-positive T cells in striatum and substantia nigra from control rats or rotenone-induced PD rats with indicated treatment. (**B**) Statistic analysis of CD4-positive T cells in the striatum area revealed a significant reduction in the SHED-CM treatment group. (**C**) Statistic analysis of CD4-positive T cells in the substantia nigra area revealed a significant reduction in the SHED-CM treatment group. The red arrows indicate stained cells. Each group *n* = 3, * *p* < 0.05, ** *p* < 0.01, ns: no significant differences.

**Figure 5 ijms-21-03807-f005:**
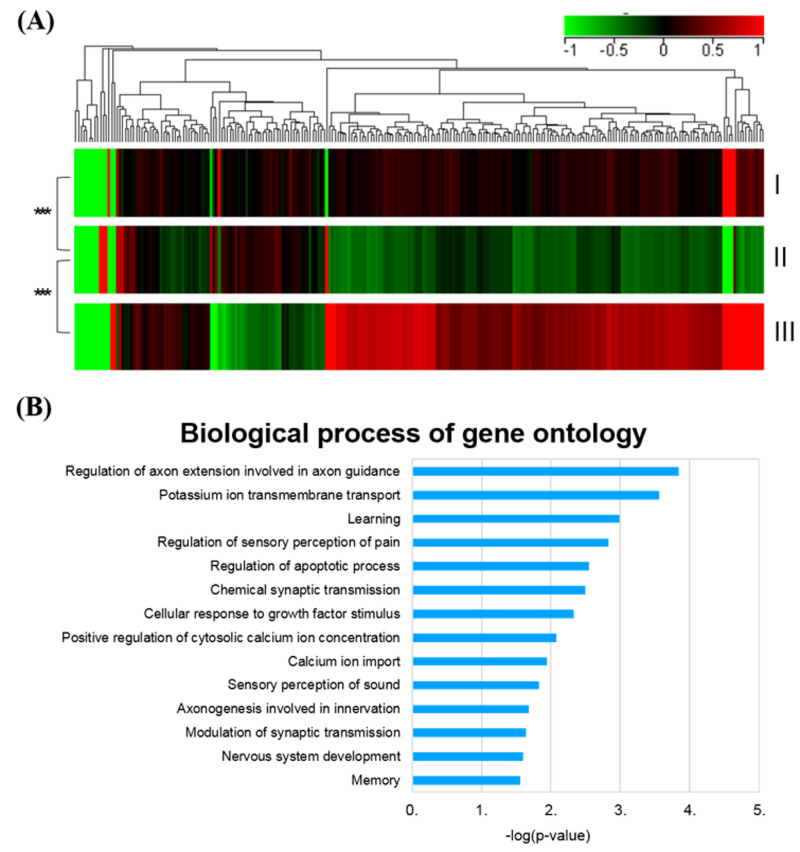
Heat-map analysis demonstrating the expression profiles of the differentially regulated genes in the rotenone-induced PD model with or without SHED-CM treatment. (**A**) Gene expression levels are presented as color variations from red (high expression) to light green (low expression) for each sample (columns) and each gene (rows). “Control” represents the untreated sham group; “Rotenone” represents a rotenone-induced PD group; “Rotenone+SHED-CM” represents the SHED-CM-treated group. *** *p* < 0.005. (**B**) Biological functions of SHED-CM upregulated genes derived from Gene Ontology (GO) enrichment of differentially expressed genes in rotenone-induced PD rats with SHED-CM treatment. Bars represent log (*p-*value) in each functional category.

**Figure 6 ijms-21-03807-f006:**
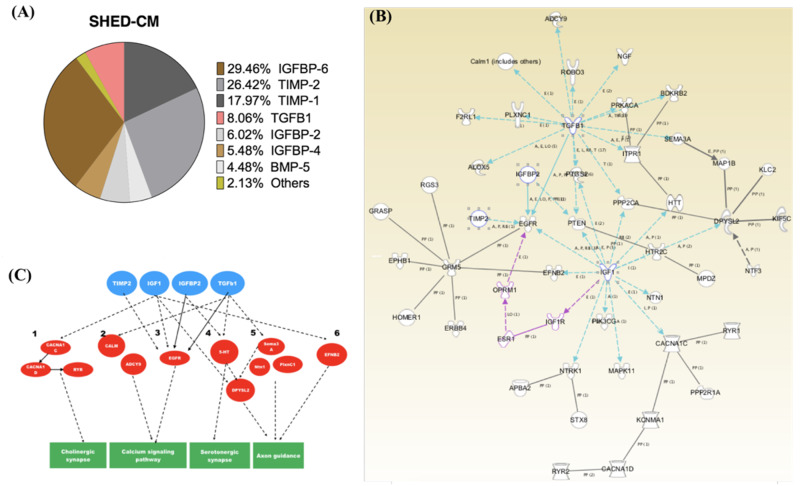
Analysis of the major constituents in SHED-CM and the plausible mechanisms of SHED-CM-mediated neuroprotective effects. (**A**) ELISA array showing the percentage of each constituent in the SHED-CM. (**B**) RNA-sequencing (RNA-seq) analysis and Ingenuity Pathway Analysis (IPA) indicate the possible mechanisms for the neuroprotective effects mediated by SHED-CM. (**C**) A simplified network depicting the possible neuroprotective mechanisms mediated by the major constituents in SHED-CM. Abbreviations: IGFBP-6, insulin-like growth factor binding protein 6; TIMP-2, tissue inhibitor of metalloproteinase 2; TIMP-1, tissue inhibitor of metalloproteinase 1; TGF-β1, transforming growth factor-beta 1, IGFBP-2, insulin-like growth factor binding protein 2; IGFBP-4, insulin-like growth factor binding protein 4; BMP-5, bone morphogenetic protein 5.

**Figure 7 ijms-21-03807-f007:**
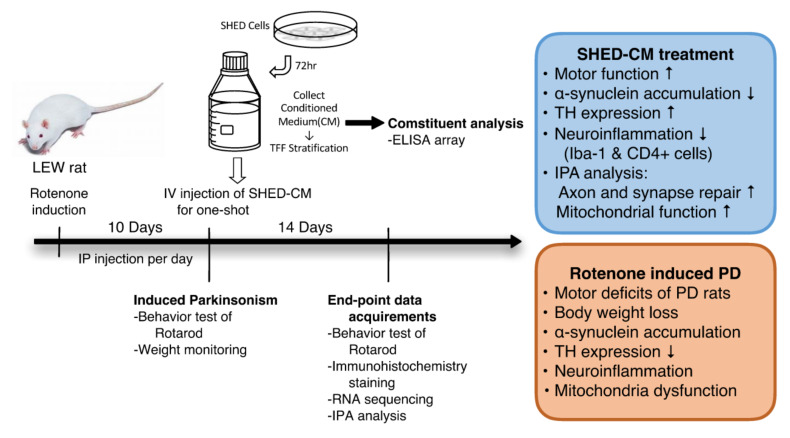
Illustrative scheme for the bioavailability of SHED-CM that can improve neuroinflammation, α-synuclein clearance, mitochondrial repair, and motor dysfunction. The rotenone-induced model exhibited motor dysfunction that recapitulates the clinical symptoms of PD. Administration of SHED-CM showed promising efficacy that improves impaired mitochondrial dysfunction, overwhelming neuroinflammation, and various neurological deficits in this PD model.

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
