# Peer review of "Improvement of Impaired Motor Functions by Human Dental Exfoliated Deciduous Teeth Stem Cell-Derived Factors in a Rat Model of Parkinson’s Disease"

_ijms, 2020, doi:10.3390/ijms21113807_

Round 1

Reviewer 1 Report

Although interesting, this study still needs considerable work.

The reviewer's main concern is experimental design of the study.

Authors induced PD with rotenone and then intravenously injected 1ml of SHED-CM (containing 100 ug of proteins). After 2 weeks functional and other tests have been performed. It remains unclear how only one intravenous injection with such a small amount of proteins could cause such a significant therapeutic effects. Authors did not discuss how biologically active proteins from SHED-CM crossed blood-brain barrier. Labeling of SHED-CM proteins and tracing them in the brain could confirm effective crossing of the BBB. Perhaps components of the SHED-CM act via peripheral mechanisms by reprogramming immune response ? All these possibilities should be explored in the future study.

There is no statistical information in Methods section and Figure legends (numbers of animals in experimental groups, etc.). No info about RNA isolation from brain tissues, no sufficient info about molecular characterization of SHED-CM. For some reason in the section about evaluation using Rotarod authors mention mouse instead of rats. 

Reviewer 2 Report

In this work, Chen et al. studied the effect of conditioned medium (CM) derived from mesenchymal stem cells of human exfoliated deciduous teeth (SHED) on motor functions, tyrosine hydroxylase (TH) and alpha-synuclein (a-syn) expression and neuroinflammation in a rat model of Parkinson’s disease (PD). The authors conclude that intravenous SHED-CM infusion improves motor deficits observed after rotenone administration, increases the expression of TH in the striatum, reduces the expression of a-syn in the striatum and substantia nigra, and reduces activation of microglia and infiltration of CD4-positive cells. They suggest that these effects are mediated by different factors present in the SHED-CM, which could be involved in calcium signalling pathways, axon guidance, mitochondrial function and synaptic modulation.

The manuscript is well-written, is easy to follow and includes interesting results. However, there are inconsistencies that need to be resolved.

Major points:

  1. In the opinion of the reviewer, outcomes should not be included in the Introduction. Moreover, other recent and relevant references should also be included (Moshy et al., 2020; Tsuruta et al., 2018).
  2. Animal models of PD based on rotenone administration can faithfully reproduce pathological hallmarks (nigral degeneration and presence of Lewy bodies) and motor deficits of the disease. However, the specificity of this compound has been questioned and some studies suggest that rotenone may produce extranigral pathology and non-motor symptoms. This model has other limitations that include a lack of reproducibility, the location and the degree of dopaminergic lesion and an important mortality rate. Although rotenone models proved to be successfully used to test novel neuroprotective strategies, authors must justify the use of this model, the route of administration and the dose selected. Why were rats anesthetized prior to intraperitoneal rotenone administration (p. 11, line 304)? Furthermore, authors include the analysis of the extend of the dopamine depletion in the striatum, but they do not include data on the degree of cell loss in the substantia nigra. Quantifications of the number of dopaminergic neurons in the substantia nigra pars compacta should be included together with the corresponding representative images. The reviewer considers these data essential in a study on neurodegeneration/neuroprotection in a neurotoxin-based experimental model of PD.
  3. Authors indicate that they use a type of mesenchymal stem cell derived from dental pulp (see Abstract and Introduction). However, they do not demonstrate the presence of stem cells in their cultures nor include any reference that supports the selected protocol. A characterization of these cultures after passage 3 (p.11, line 327) showing the multipotency and the mesenchymal fate of these cells must be included.
  4. An explanation of why they collected from SHED-CM the fraction of proteins with molecular weights between 5-30 KDa and why they chose 30μg/ml and 100μg/ml as starting doses of CM must be included.
  5. Information about tissue fixation method, cryoprotection as well as cutting and thickness of sections should be mentioned. Moreover, information about animals where primary and secondary antibodies were obtained, and the meaning of ABC abbreviation and its commercial supplier should be added.
  6. A detailed explanation of the method followed to determine intensity of TH and a-syn labelling and the system of quantification of Iba-1 and CD4 cells should be included.
  7. The tissue from which RNA was obtained and the method for RNA isolation are not mentioned.
  8. In Fig 1A, a total of 27 rotenone-treated rats were included. However, analysis of rotaroad behavioural test (Fig. 1B) includes only 6 rotenone-treated rats. Please, provide an explanation for this difference. Moreover, authors show a moderate loss of weight in rotenone-treated group, but they do not indicate whether this loss is or not significant. In Fig. 1C, authors indicate that ventriculomegaly with remarkable shrinkage of the whole striatum and severe hydrocephalus was noted in rotenone-treated rats and suggest that these phenotypes validate the induction of PD in these rats. This must be clarified. In Fig. 1D, results showed that treatment with 30μg and 100μg SHED-CM restores the ability to perform rotaroad test in rotenone-treated animals with respect to animals treated with rotenone alone. However, comparisons between control group and rats treated with rotenone + different doses of SHED-CM are not represented. These comparisons (Fig. 1D and, also in the corresponding histograms in Fig. 2, 3 and 4) could provide additional information.
  9. Authors do not explain why the study of activation of microglial cells is relevant (i.e. cells positive for Iba-1) in cerebral cortex. If they consider that this information is significant, data about the expression of CD4 cells in this region should also be included.
  10. Authors suggest that SHED-CM mediates neuroprotective effects via different mechanisms (modulation of cholinergic/serotonergic synapses, axon guidance, recovery of mitochondrial function, reduction of neuroinflammation, a-syn clearance, etc.). However, the mechanisms referred are not always the same in the different sections of the manuscript (see Abstract, Results, Fig. 6, Discussion and Conclusion). Most importantly, no experimental evidence is provided to support some of them. Mechanisms proposed and the overall conclusions obtained from the study should be in accordance and must be matched in all sections.
  11. Indeed, as the authors demonstrate, mesenchymal stem cells obtained from dental pulp produce many bioactive factors with neuroprotective and anti-inflammatory effects. It is also well-known that many of these factors are growth factors and many of them elicit potent neuroprotective effects on dopaminergic neurons. Although most of the common neurotrophic factors have high molecular weight, some of them have molecular weights that would fall within the fraction collected in this work. These aspects should be taken into consideration in the Discussion. In addition, previous works in which these cells are used in animal models of PD should be cited (Fujii et al., 2015; Jarmalaviciute et al., 2015; Narbute et al., 2019; Nesti et al., 2011; Nosrat et al., 2004; Zhang et al., 2018).

Minor points:

  1. In the title, it would be more appropriate ‘in a rat model of Parkinson’s disease’ than ‘in the rat model of Parkinson’s disease’.
  2. In Abstract, information about the route of SHED-CM administration should be included.
  3. Please, review the sentence ‘It was plausible that the efficacy attributed to the bioactive constituents secreted from SHED.’ (p. 1, line 74).
  4. Change ‘mouse’ (p. 12, line 351) for ‘rat’ in section ‘Behavior evaluation by rotaroad system’.
  5. Please, indicate the statistical program used.
  6. Abbreviations must be used in the same way (a-syn and ASN are used for a-synuclein). Moreover, they should be explained only in their first use and if an abbreviation is mentioned, it should be used later on in the manuscript. Table of abbreviations is not complete and should be revised.
  7. The manuscript should be carefully checked to avoid typos (‘substantial nigra’ instead of ‘substantia nigra’, ‘human exfoliated deciduous teeth (SHED) ‘cell’ isolation and culture, neuro-inflammation/neuroinflammation).

Reviewer 3 Report

The manuscript by Chen et al. evaluated the therapeutic effect of SHED-derived conditioned medium (SHED-CM) in a rotenone-induced PD rat model, showing that administration of SHED-CM exerted different beneficial actions. Gene ontology analysis revealed that the biological process affected by SHED-CM were implicated in neurodevelopment and nerve regeneration and authors examined the major constituents of SHED-CM.

Stem cells seem to be promising as new therapeutic approaches for different diseases, but stem-cell free therapy, using their derivatives such as conditioned medium, has emerged as a potentially safer and cost-effective alternative. Although the work may be interesting, there are some major points to address.

My major concerns are provided below.

In particular, the materials and methods section and results about RNA sequencing and network analysis are not well written and confusing. The major concerns are the following:

-The authors indicated they used as reference genome Ensembl GRCh38, but this is a human genome. The alignment must be performed on rat genome.

- Why did the authors use Cuffdiff? There are new tools such as DESeq2 or EdgeR.

- In the RNA sequencing paragraph in the materials and methods section specify from which tissue the RNA was extracted and the kit used, the Illumina kit and Illumina instrument used, the number of replicate.

- For GO analysis there is a confusion between biological process and function and it is not clear if only upregulated genes were considered. It is not clear the differentially expressed genes to which comparison are referred to: rotenone-induced PD rats with SHED-CM treatment compared to?

In summary, all the parts referred to RNA sequencing need to be deeply revised.

Statistical analysis was performed using Student t-test, but t-tests can compare two groups. When more than 2 groups are present, it is more appropriate to use ANOVA or non-parametric Kruskal–Wallis test followed by post hoc test. However, at first it is necessary to verify normal distribution of the data to choose between parametric and non parametric test.

Characterization of SHED should be shown.

Why in the preparation of CM only the fraction with molecular weight between 5-30 kDa was collected and used for the experiments? Explain also how the tested CM doses (30 and 100 µg/ml) were chosen.

Did the authors isolated striatum and substantia nigra? In the results, but not in materials and methods section, the authors specified that striatum and substantia nigra were isolated and collected for the analysis.

To determine the constituents of SHED-CM the authors indicated in the results that they performed ELISA. This part is completely missing in the materials and methods section. A better characterization of CM should be performed. At the moment, authors indicated only percentage.

In figure 1 B the statistical significance is expressed with ****, but it is missing in the figure legend.

Figure 4: specify in figure legend what the red arrows indicate.

Round 2

Reviewer 1 Report

I recommend to radically redesign this study. You should definitely demonstrate that factors from conditioned medium penetrate BBB and enter into the CNS in sufficient quantities. Alternatively, I would suggest to pay attention at the immunomodulatory action of your therapeutic proteins. 

Reviewer 2 Report

In the revised version of the paper the authors have properly considered the reviewer’s comments and have incorporated most of the suggested modifications with the corresponding explanations.

The reviewer has just some minor comments before the manuscript can be considered for publication:

1) In the new figure 1, some histograms are overlapping.

2) The description of the method followed to determine the optical density/intensity of labelling could be useful for readers and should be properly included in the manuscript.

“And then, we used ImageJ software for analyzing the IHC images and quantification. In single labeling IHC, the stained cells that should be "darker" than the background, we switched the IHC images to grayscale and defined the grey threshold, then measured the gray integrated density of the staining above the threshold. The definition of the threshold was the gray level value when the stained cells were completely removed, leaving only the background.”

3) In the new version of the manuscript, authors included information about the effect of 400ug/ml of SHED-CM-treatment on rotarod performance test (p. 3, lines 114-115). Consequently, this dose must be mentioned in methods.

4) Authors are absolutely right that ‘substantia nigra’ is the correct term. This is what the reviewer meant to say. I apologize for the misunderstanding. However, there are still some typos that need to be corrected (see for example in Methods: 10mm, 60C, different style and font size, etc.).

5) Please, change in page 18, line 410 ‘Avidin-Biotin complex’ for ABC.

Reviewer 3 Report

In the revised manuscript, the authors answered to comments. However, there are some comments that should be addressed before publication.

- Correct in Figure 5 "Biology process" with "Biological process". In the abstract the authors stated "Gene ontology analysis indicated that the molecular function of genes affected by SHED-CM was primarily implicated in neurodevelopment and nerve regeneration.", while in figure 5 they wrote biological process. Then, I think that in the abstract molecular function should be changed with biological process.

- The Illumina kit used for RNA sequencing and the Illumina instruments used should be indicated.

- I suggest to include in the manuscript the explanation for the preparation of the fraction 5-30 kDa of the CM, as the authors explained in the point-to-point response. It can be helpful for the readers. I appreciated that the authors included in the manuscript that the 400μg/ml of SHED-CM treatment did not reveal significant improvement than the group of 100μg/ml treatment. However, I suggest to include it in the figure. I also suggest to indicate in the manuscript that the doses 10 and 30 µg were not efficacious as the authors explained in the reviewer response.

- Correct the paragraph 2.2 in accordance to the new figure 2. In particular, in the paragraph the TH results in substantia nigra should be added and correct the letters in line with the figure.

- Pay attention to the abbreviations in the discussion section, some terms were already abbreviated in the previous paragraphs.

- In the paragraph regarding rotenone-induced Parkinson's disease in rats authors indicated that a total of 30 rats were used. Explain their division in the groups.

- In the materials and methods section, in the paragraph SHED-CM treatment, I suggest to indicate the number of animals for each group. Moreover, in figure 1D also the groups treated only with SHED-CM were indicated and I suggest to include them in the materials and methods section.

- In the paragraph regarding immunohistochemical staining, line 400, is 10 mm thick correct?

- In the title of the paragraph 2.1 correct "SHEM-CM" with "SHED-CM". In the same paragraph (line 99) correct "theses rats" with "these rats".
